# Correlation between Blunted Nocturnal Decrease in Diastolic Blood Pressure and Oxidative Stress: An Observational Study

**DOI:** 10.3390/antiox11122430

**Published:** 2022-12-09

**Authors:** Nestor Vazquez-Agra, Anton Cruces-Sande, Estefania Mendez-Alvarez, Ramon Soto-Otero, Sergio Cinza-Sanjurjo, Jose-Enrique Lopez-Paz, Antonio Pose-Reino, Alvaro Hermida-Ameijeiras

**Affiliations:** 1Division of Hypertension and Cardiovascular Risk, Department of Internal Medicine, University Hospital of Santiago de Compostela, 15706 A Coruña, Spain; 2Department of Psychiatry, Radiology, Public Health, Nursing and Medicine, University of Santiago de Compostela, 15782 A Coruña, Spain; 3Laboratory of Neurochemistry, Department of Biochemistry and Molecular Biology, Faculty of Medicine, University of Santiago de Compostela, 15782 A Coruña, Spain; 4Department of Family and Community Medicine, Porto do Son Health Center, 15970 A Coruña, Spain; 5Biomedical Research Center on Cardiovascular Diseases Network (CIBERCV), Health Institute Carlos III (ISCIII), 28029 Madrid, Spain

**Keywords:** blood pressure, arterial hypertension, circadian profile, dipper, oxidative stress, thiobarbituric acid reactive substances, reduced thiols

## Abstract

An impaired nocturnal decrease in diastolic blood pressure (DBP) increases the blood pressure (BP) load, which is a main factor in endothelial dysfunction, atherosclerosis, and arterial stiffness. We aimed to quantify some markers of oxidative stress in hypertensive patients, to compare their levels between individuals with dipper and non-dipper DBP profiles, and to assess their correlation with the nocturnal DBP (nDBP) dipping. It was an observational study that included patients older than 18 years with a diagnosis of essential hypertension who consented to participate. The collected variables were some indices of 24-h ambulatory blood pressure monitoring, demographic, epidemiological, clinical, and laboratory variables. Plasma thiobarbituric acid reactive substances (TBARS) and reduced thiols, together with serum vitamin E, vitamin A, copper (Cu), and zinc (Zn) levels were assessed as oxidative stress markers. We recruited 248 patients with a median age of 56 years (56% women). The percentage of nDBP dipping showed a weak positive correlation with reduced thiol, vitamin E, and vitamin A levels; and a weak negative correlation with Cu levels. We also found a negative correlation between nDBP dipping and the TBARS/Thiol, TBARS/Vitamin E, and TBARS/Vitamin A ratios. After multivariate analysis, we found that increased TBARS/Thiol ratio and serum Cu levels were associated with a higher risk of a non-dipper DBP profile. As in other situations of increased cardiovascular risk, an impaired nDBP decrease may coincide with abnormalities in redox status.

## 1. Introduction

Arterial hypertension (AHT) is an important cause of morbidity and mortality worldwide. Blood pressure (BP) levels follow a circadian pattern characterized by a physiological nocturnal decrease. The 24-hour ambulatory BP monitoring (24h-ABPM) indices are better predictors of cardiovascular risk (CVR) than isolated BP measurements and night-time BP is more correlated with cardiovascular disease (CVD) than daytime BP [1,2,3].While systolic blood pressure (SBP) is the main determinant of adverse events in older patients, diastolic blood pressure (DBP) levels are more harmful in individuals younger than 50 years [4]. The presence of a pro-inflammatory milieu and redox imbalance have been linked to short-term abnormalities in vascular tone that could underlie high DBP levels in young and middle-aged patients [5]. Additionally, some cellular and physiological models have pointed to redox imbalance and unfavorable inflammatory status as factors correlated with increased atherosclerosis (AS) burden and arterial stiffness in middle-aged and elderly hypertensive patients [6].

Multiple cellular processes such as enzymatic activity, molecular autoxidation, mitochondrial electron transport, microsomal reactions, and those involving divalent metal scavengers and hemoproteins are sources of free radicals (FR), highlighting the superoxide anion (O_2_^−^) [7,8]. Some correlated chemical reactions that are mediated by divalent cations such as copper (Cu) and iron (Fe) lead to the formation of reactive oxygen species (ROS) while zinc (Zn) exerts a protective effect [9]. The hydroxyl radical (^•^OH), because of its high oxidation potential and reactivity, is the most concerning one [10]. 

ROS, particularly ^•^OH, attack polyunsaturated fatty acids given rise to intermediate lipid peroxides that propagate the oxidative process to other cellular elements in a chain reaction. Intermediate and final lipid peroxidation products such as malondialdehyde (MDA) and 4-hydroxynonenal (4-HNE) are extremely toxic to cell membranes [11]. In a pro-oxidant environment, carboxyl groups of proteins are oxidized to ketones, while reduced thiols, which are an important extracellular antioxidant defense, are oxidized to disulfide bonds and other compounds. Excess carbonyl groups and thiol/disulfide imbalance are related to protein structural alterations and enzyme dysfunction. Lipid and protein oxidation are involved in several clinical acute and chronic clinical entities including AS related disorders [12,13,14].

Fortunately, our internal milieu has intrinsic antioxidant defenses while we regularly ingest multiple dietary antioxidants, such as vitamins C, A, E, and some phenols [15].

Some reviews have pointed to a possible role of redox imbalance in BP levels [16,17]. However, the impact of oxidative stress on the circadian blood pressure profile is poorly understood and even less is known about its influence on nocturnal DBP (nDBP) indices. Thus, we aimed to quantify some markers of oxidative stress in hypertensive patients, to compare their levels between patients with dipper and non-dipper DBP profiles and to assess their correlation with the nDBP dipping.

## 2. Materials and Methods

### 2.1. Study Design and Patients

This was an observational study conducted between January and June 2022 in the Arterial Hypertension and Cardiovascular Risk Unit belonging to the Internal Medicine Department of the Santiago de Compostela Hospital (Galicia, Spain).

We recruited patients with essential hypertension and older than 18 years who consented to participate. Individuals with a clinical suspicion or definitive diagnosis of secondary AHT were excluded. The presence of actual tobacco use, heavy alcohol drinking, diabetes mellitus (DM), established cardiovascular (CVD), respiratory, or renal disease were exclusion criteria, and the definitions were considered according to the current literature [18,19,20,21].

### 2.2. Parameters of ABPM Collection

Patients underwent 24-hour ABPM using an electronic (oscillometric) upper-arm cuff device (Space-Labs 90207^®^ device; Space-Labs Inc., Redmond, WA, USA) validated according to the STRIDE blood pressure protocol that is recommended by the European Society of Arterial Hypertension (ESH). The procedure was explained to the patients by providing a form to record sleeping times, drug intake, or any problems during the recording. We followed the recommendations endorsed by the ESH for out-of-office BP measurement, which are summarized as follows: performance of the test on a working day; frequency of measurement with a minimum of 20 and 30 min during day and night, respectively; use of the non-dominant arm, with center bladder over the brachial artery; appropriate cuff size to fit the individual’s arm circumference; and evaluation of day- and night-time periods only according to patient’s report [22].

The following indices were available from the 24 h ABPM recordings: Average 24-hour SBP (24-hSBP) and DBP (24-hDBP), average daytime SBP (dSBP) and DBP (dDBP), average nocturnal SBP (nSBP) and DBP (nDBP), and percentage of nSBP and nDBP dipping. A non-dipper DBP profile was defined as a decrease in nDBP of lower than 10% compared with the average daytime values. A dipper DBP pattern was defined as a decrease in nDBP equal to or higher than 10% compared with the average daytime values. ABPM assessment was considered reliable if more than 70% expected measurements were valid [2,22].

### 2.3. Clinical and Laboratory Variables

We collected information about age, sex, alcohol consumption (no/yes), and former tobacco use (no/yes). Body weight and height were measured in kilograms (kg) and meters (m), respectively. Body mass index (BMI) was calculated as the ratio of weight to height squared, measured in kg/m^2^ [23]. Waist circumference (WC) was assessed above both iliac crests using a qualified tape and measured in centimeters (cm) [24]. The use of antihypertensive drugs and the degree of therapeutic compliance assessed by the Morisky–Green questionnaire were considered [25].

Blood samples were obtained at 08:00 AM following overnight fasting of 12 h and ensuring a minimum of 12 h since the last use of antihypertensive drugs. The included variables were fasting plasma glucose (FPG), uric acid, creatinine, total cholesterol (TC), and triglyceride (TG) levels. Concentrations were given in mg/dL. Total protein levels were expressed in g/dL.

Cu and Zn levels were measured using atomic absorption spectrometry and levels were expressed in µg/dL while vitamin E and vitamin A levels were measured using high-performance liquid chromatography (HPLC). Given the influence of serum lipids on vitamin E and vitamin A levels, we adjusted them by TG concentrations. Levels were expressed in µg per mg (µg/mg) of serum TG [26,27].

### 2.4. Lipid and Protein Oxidation: Assessment of Thiobarbituric Acid Reactive Substances (TBARS) and Reduced Thiols

Blood samples were collected in tubes with EDTA, centrifuged in less than 1 h after extraction at 1000 G and 4 °C for 10 min. The plasma fraction was deposited at −80 °C for less than 1 month until analysis. All samples were analyzed in quadruplicate and a standard calibration for each protocol was performed to obtain a linear model with a coefficient of determination (R^2^) greater than 98% [28].

Thiobarbituric acid (TBA) forms an adduct with MDA and other lipid peroxidation products under high temperature (90–100 °C) and acidic conditions yielding a violaceous pigment that can be measured spectrophotometrically at 530–540 nm. The absorbance is directly proportional to the level of plasma lipid peroxides. We followed the protocol of Ohkawa et al. and the concentrations were expressed in nanomoles (nmol) per mg (nmol/mg) of TC [29,30].

The assessment of reduced thiols is a well-systematized technique for the quantification of protein oxidation. The concentration of reduced thiols is inversely proportional to the level of protein oxidation in the sample. Ellman’s technique uses 5,5-dithio-bis-(2-nitrobenzoic acid) as a reagent to form a compound with the sulfhydryl groups of some amino acid residues in proteins, yielding a colorful pigment that can be measured spectrophotometrically at 412 nm and whose absorbance is proportional to reduced thiol levels in plasma proteins. The concentrations were expressed in micromoles (µmol) per mg (µmol/mg) of plasma proteins or lipoproteins when there were differences in total protein or lipoprotein levels between the comparison groups [31,32].

### 2.5. Ethics Statement

This study was conducted in accordance with the ethical principles of the Declaration of Helsinki and standards of good practice (NBP) in research of the Galician (Spain) health service (SERGAS). All patients who consented to participate provided written informed consent. The Research Ethics Committee of Santiago-Lugo approved the study protocol (code number 2021/401).

### 2.6. Statistical Analysis

Statistical analysis was developed using SPSS 22.0 statistical software (SPSS Inc, Chicago, IL, USA). The sample size calculation was performed for a statistical power greater than 0.8. A *p*-value lower than 0.05 (*p* < 0.05) was considered for statistical significance. We performed a descriptive analysis in which the frequencies of qualitative variables were expressed as number (n) and percentage (%). The Kolmogorov–Smirnov test was used to determine whether quantitative continuous variables were normally distributed. Normally distributed variables were expressed as mean (m) and standard deviation (SD) and non-normally distributed variables were expressed as median and interquartile range (IQR).

We conducted a comparative univariate analysis between patients with dipper and non-dipper DBP profiles. The chi-square test was used to compare categorical variables, while quantitative variables were compared using the Student’s *t*-test or Mann–Whitney U test where appropriate. A linear correlation analysis between nDBP dipping and TBARS, reduced thiols, vitamin E, vitamin A, Cu levels, and Cu/Zn ratio was performed. We also explored the relationship between some of the oxidant and antioxidant variables to perform a linear correlation analysis of the TBARS/thiol, TBARS/Vitamin A, and TBARS/Vitamin E ratios with the nDBP decrease. The Pearson or Spearman correlation coefficients were used as appropriate.

We developed a binary logistic regression model for the risk of a non-dipper DBP profile based on each redox marker to evaluate confounding-interaction phenomena between variables and to reduce the number of variables in the final model. The redox markers that maintained statistical significance, and the clinical and laboratory variables that showed differences between the comparison groups were included in a global logistic regression model for the risk of a non-dipper DBP profile.

## 3. Results

We recruited 248 patients with a mean age of 56 years, of whom 138 (56%) were women. Approximately one in five and one in three individuals had a low alcohol consumption or were former smokers, respectively. Renin–angiotensin–aldosterone system (RAAS) blockers were the most commonly used antihypertensive drugs. Table 1 shows in detail the clinical and laboratory findings.

### 3.1. Differences between Patients with Dipper and Non-Dipper DBP Profiles

Patients with dipper and non-dipper DBP profiles showed clinically relevant differences in age, alcohol intake, WC, BMI, 24h-SBP, nSBP, nDBP, and diuretic use. The degree of therapeutic compliance was similar in both groups. Regarding laboratory variables, patients with dipper and non-dipper DBP patterns showed differences in serum TG levels (Table 1).

We found relevant differences (*p* < 0.05) in the median levels of vitamin E, vitamin A, TBARS, and reduced thiols between hypertensive patients with dipper and non-dipper DBP profiles. The results showed lower levels of reduced thiols, vitamin A, and vitamin E with higher concentrations of TBARS in patients with a non-dipper DBP pattern. We found no relevant results on Cu and Zn levels. The results are shown in Figure 1.

### 3.2. Correlations between nDBP Dipping and Oxidative Stress Markers

We found a weak positive correlation between the percentage of nDBP dipping and the levels of reduced thiols (Rho = 0.144, *p* = 0.023), vitamin E (Rho = 0.147, *p* = 0.023), vitamin A (Rho = 0.156, *p* = 0.016), and a weak negative correlation with Cu (Rho = −0.127, *p* = 0.046) levels. We found no relevant results on TBARS levels and the Cu/Zn ratio for the nDBP decrease. Scatter plots in Figure 2 show the correlations between the redox markers and nDBP dipping.

When exploring the relationship between some oxidant and antioxidant variables, we found a weak negative correlation between nDBP dipping and the TBARS/Thiol (Rho = −0.185, *p* = 0.004), TBARS/Vitamin E (Rho = −0.165, *p* = 0.012), and TBARS/Vitamin A (Rho = −0.195, *p* = 0.003) ratios. The results are shown in Figure 3.

### 3.3. Multivariate Analysis: Binary logistic Regression Models for the Presence of a Non-Dipper DBP Profile

Taking into account the differences in age, BMI, WC, alcohol intake, 24h-SBP, nSBP, nDBP, diuretic use, and TG levels between patients without and with a non-dipper DBP profile, and testing the possible interaction-confusion phenomena, we constructed binary logistic regression models for each redox marker. TBARS (B = 0.738, *p* = 0.002, Exp (B) = 2.091, 95%CI = 1.319–3.316), reduced thiols (B = −1.772, *p* = 0.042, Exp (B) = 0.170 95%CI = 0. 031–0.939), vitamin A (B = −1.225, *p* = 0.016, Exp (B) = 0.294 95%CI = 0.109–0.792), and Cu (B = 0.017, *p* = 0.049, Exp (B) = 1.017 95%CI = 1.001–1.035) levels; and the TBARS/Thiol (B = 0.475, *p*< 0.001, Exp (B) = 1.609, 95%CI = 1.246–2.075), TBARS/Vitamin E (B = 0.135, *p* = 0.002, Exp (B) = 1.144, 95%CI = 1.051–1.246), and TBARS/Vitamin A (B = 0.896, *p*< 0.001, Exp (B) = 2.450, 95%CI = 1.578–3.805) ratios were relevant variables within their respective models. The results are provided in Appendix A.

The final and joint binary logistic regression model for clinical, laboratory, and redox markers with relevant results showed that increased TBARS/Thiol ratio and serum Cu levels were associated with a higher risk of a non-dipper DBP profile. The findings are summarized in Table 2.

## 4. Discussion

The results are summarized as follows: (1) The percentage of nDBP dipping showed a positive correlation with reduced thiol, vitamin E, and vitamin A levels; and a negative correlation with Cu levels. (2) We also found a negative correlation between nDBP dipping and the TBARS/Thiol, TBARS/Vitamin E, and TBARS/Vitamin A ratios. (3) After multivariate analysis, we found that TBARS/Thiol ratio and serum Cu levels were associated with the risk of a non-dipper DBP profile.

ROS enhance some cellular pathways related to inflammation, highlighting their role in the activation of the NF-KB protein complex [33]. The presence of an unfavorable inflammatory and redox status, arterial wall tension, and turbulent blood flow are factors involved in endothelial dysfunction (ED). ED impairs the arterial wall smooth muscle relaxation leading to inadequate lowering of DBP levels. The same variables that lead to ED and impaired DBP also contribute to the long-term development of AS and arterial stiffness [34,35].

High serum Cu levels were congruent with the presence of an impaired nDBP dipping. Cu, by oxidation number change, can react with O_2_^−^ to produce H_2_O_2_ by the Haber–Weiss reaction. The Fenton reaction between Cu(I), among other reduced divalent metals, and H_2_O_2_ to produce ^•^OH is also thermodynamically favorable. Zn has a stable oxidation number and is capable of displacing other divalent metals from their binding sites and inhibiting Cu-dependent oxidative damage [36,37].

We found that patients with a non-dipper DBP profile had higher plasma levels of TBARS than dipper individuals. Lipid peroxidation is an important mechanism of oxidative stress mediated damage and some divalent metals such as Cu and Fe can amplify this phenomenon [38]. The excess of lipid peroxides impairs fluidity and permeability of cell membranes and induces structural and functional damage to transmembrane proteins. Some lipid peroxidation end byproducts such as MDA and 4-HNE have been linked to mitochondrial and genetic damage, cell death, neurodegenerative processes, and inflammation. Patients with high CVR and AS burden, including hypertensive individuals, have also shown high levels of lipid peroxides [39].

Although TBARS levels did not show a linear correlation with nDBP dipping, we found that patients with non-dipper DBP profile had lower levels of reduced thiols, vitamin E, and vitamin A, while nDBP dipping was positively correlated with the levels of these antioxidants. This phenomenon could reflect antioxidant pool depletion as these molecules are some of the most important barriers against oxidant processes such as lipid peroxidation [40].

The results suggest that the TBARS/antioxidant ratios could be at least as relevant as their absolute levels in predicting an impaired nDBP decrease. Redox status is a dynamic phenomenon consisting of numerous and continuous chemical reactions so that redox imbalance could appear as an elevation of oxidative stress markers, decrease of antioxidant molecules, or abnormal oxidant/antioxidant ratios [41].

The thiol pool represents the main immediate antioxidant line of defense in plasma and the cysteine-rich residues of plasma proteins, particularly albumin, are the main source of them. Reduced thiols maintain a critical redox balance that can be affected by the action of some reactive species and transition metals. However, under physiological conditions, the glutathione metabolic pathway ensures adequate replenishment of reduced thiol levels [42,43]. Abnormalities in the homeostasis of reduced thiols cause structural and functional damage to proteins and several clinical situations of increased CVR have shown lower levels of plasma reduced thiols [44].

Vitamin E has been proposed as the most important inhibitor of lipid peroxidation in vivo because of its great capacity to sequester lipid peroxyl radicals, thus preventing their reaction with nearby fatty acids or membrane proteins. Vitamin C cooperates with vitamin E by regenerating alpha-tocopherol in membranes and lipoproteins. Levels of vitamin A and its derivatives can also inhibit lipid peroxidation processes, although their action depends on some chemical conditions such as oxygen concentration [12,45,46].

From a clinical perspective, most of the results agree with those found in other scenarios of increased CVR, such as the presence of AHT, DM, hyperlipemia (HLP) and some types of established CVD [47,48]. Several studies in AHT have shown that some of the assessed redox markers correlate with SBP and DBP levels [49,50]. Far fewer studies suggested that dipper and non-dipper hypertensive patients show different redox profiles than normotensive patients [51,52].

### Limitations and Strengths

This was an observational, single-center study that was conducted in actual clinical practice. We predominantly included young and middle-aged, white, non-smoking individuals without CVD, so the results should be interpreted with caution when applying them to other populations. The classic distinction between dipper and non-dipper patients is a more complex reality nowadays with the presence of some other circadian BP patterns with their own peculiarities as the riser or very dipper BP profiles [22]. A large number of variables could influence BP profile or levels of redox markers; thus, it is possible that any relevant factor is excluded from the models.

Laboratory tests were performed at one point and under similar conditions for all patients. Although colorimetric techniques are feasible and sensible, they tend to be more non-specific than other methods such as chromatography [53]. Additionally, the assessment of a wide variety of redox markers available in the CVR literature could have increased the robustness of the results. Despite the multivariate analysis, some differences between the comparison groups may have influenced several results. Differences in the lipid profile could affect the validity of results in some redox markers, so concentrations were expressed as a function of serum lipoprotein levels [30,32].

## 5. Conclusions

Insufficient nocturnal decrease in diastolic blood pressure increases the high blood pressure load, which is a main factor in endothelial dysfunction, atherosclerosis, and arterial stiffness. A stiff arterial wall is more sensitive to the damage induced by elevated SBP in middle-aged and elderly hypertensive patients, leading to worse cardiovascular outcomes. Therefore, the evaluation of factors related to impaired nDBP dipping could be relevant and oxidative stress may be playing a role, although more studies are needed. The measurement of some markers of oxidative stress might be useful to take decisions and make recommendations based on the levels of nDBP and the percentage of nDBP dipping.

## Figures and Tables

**Figure 1 antioxidants-11-02430-f001:**
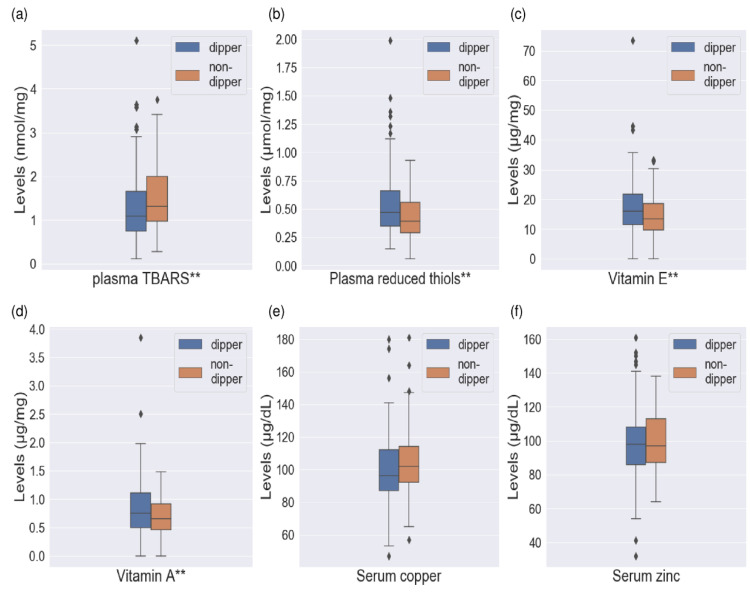
Levels of oxidative stress markers in hypertensive patients with dipper and non-dipper DBP profiles. Box-and-whisker diagram. The horizontal line of the box represents the median and the upper and lower edges represent the 3rd and 1st quartiles (Q), respectively. The edges of the upper and lower whiskers represent the Q3 + 1.5 × interquartile range and Q1 − 1.5 × interquartile range, respectively. The data represented by diamonds are the outliers. (**a**) Plasma TBARS (nmol/mg). Dipper: 1.08 (0.9), non-dipper: 1.31 (1.0), *p* = 0.005; (**b**) plasma reduced thiols (µmol/mg). Dipper: 0.47 (0.3), non-dipper: 0.39 (0.3), *p* = 0.019; (**c**) vitamin E (µg/mg). Dipper: 16.1 (10.1), non-dipper: 13.4 (9.3), *p* = 0.029; (**d**) Vitamin A (µg/mg). Dipper: 0.75 (0.62), *p* = 0.032; non-dipper: 0.64 (0.50); (**e**) serum copper (µg/dL). Dipper: 96.5 (26), non-dipper: 102.0 (24), *p* = 0.102; (**f**) serum zinc (µg/dL). Dipper: 98.0 (22), non-dipper: 97.0 (26), *p* = 0.550. TBARS—thiobarbituric acid reactive substances. ** Refers to *p*-value lower than 0.05. Results are expressed in median and interquartile range.

**Figure 2 antioxidants-11-02430-f002:**
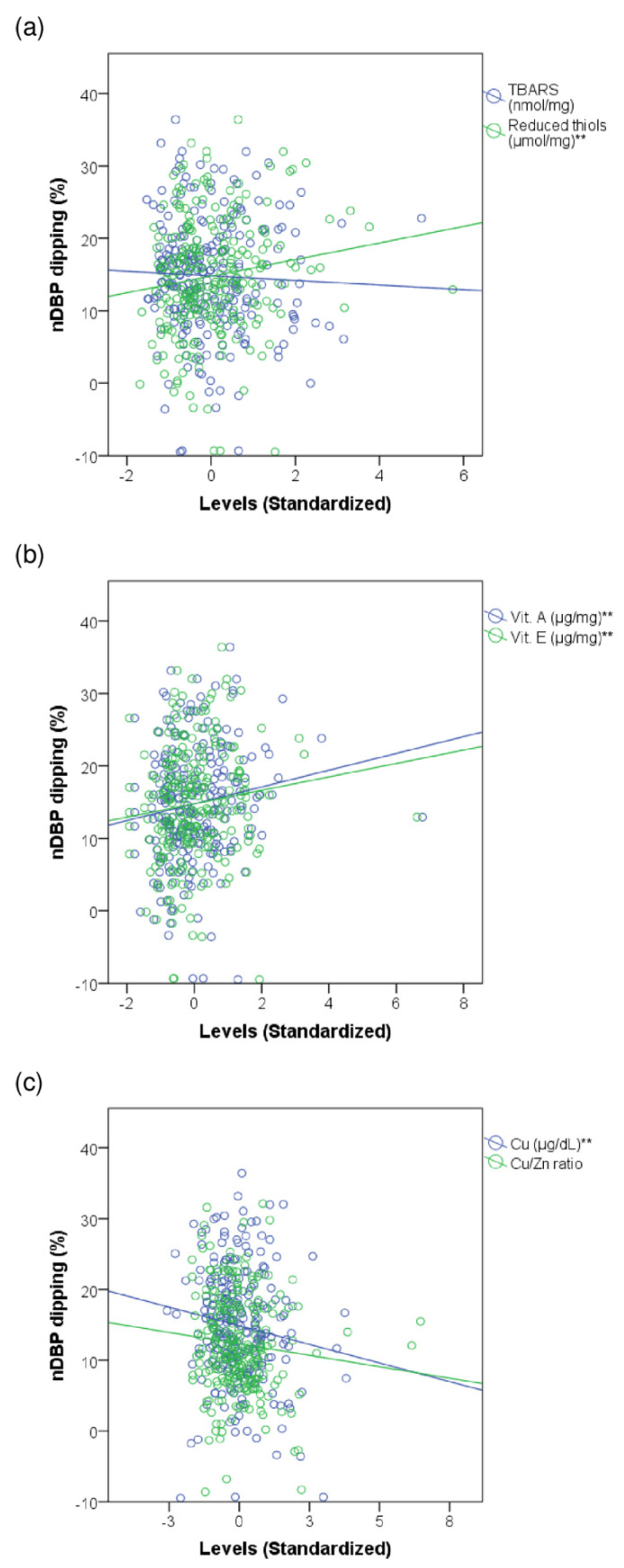
Correlations between nocturnal blood pressure dipping and some markers of oxidative stress. The scatter plots represent the degree of nDBP dipping according to the levels of TBARS, reduced thiol, vitamin E, vitamin A, Cu, and Cu/Zn ratio. Linear correlation equations (standardized coefficients). (**a**) TBARS: nDBP dipping (%) = 14.82 + (−0.32) × TBARS. Reduced thiols: nDBP dipping (%) = 14.79 + 1.14 × reduced thiols; (**b**) Vit E: nDBP dipping (%) = 14.78 + 0.93 × Vit E. Vit A: nDBP dipping (%) = 14.78 + 1.16 × Vit A; (**c**) Cu: nDBP dipping (%) = 14.88 + (−1.05) × Cu. Cu/Zn ratio: nDBP dipping (mmHg) = 12.32 + (−0.65) × Cu/Zn; nDBP—nocturnal diastolic blood pressure. TBARS—thiobarbituric acid reactive substances. Vit—vitamin. Cu—copper. Zn—zinc. ** Refers to *p*-value lower than 0.05.

**Figure 3 antioxidants-11-02430-f003:**
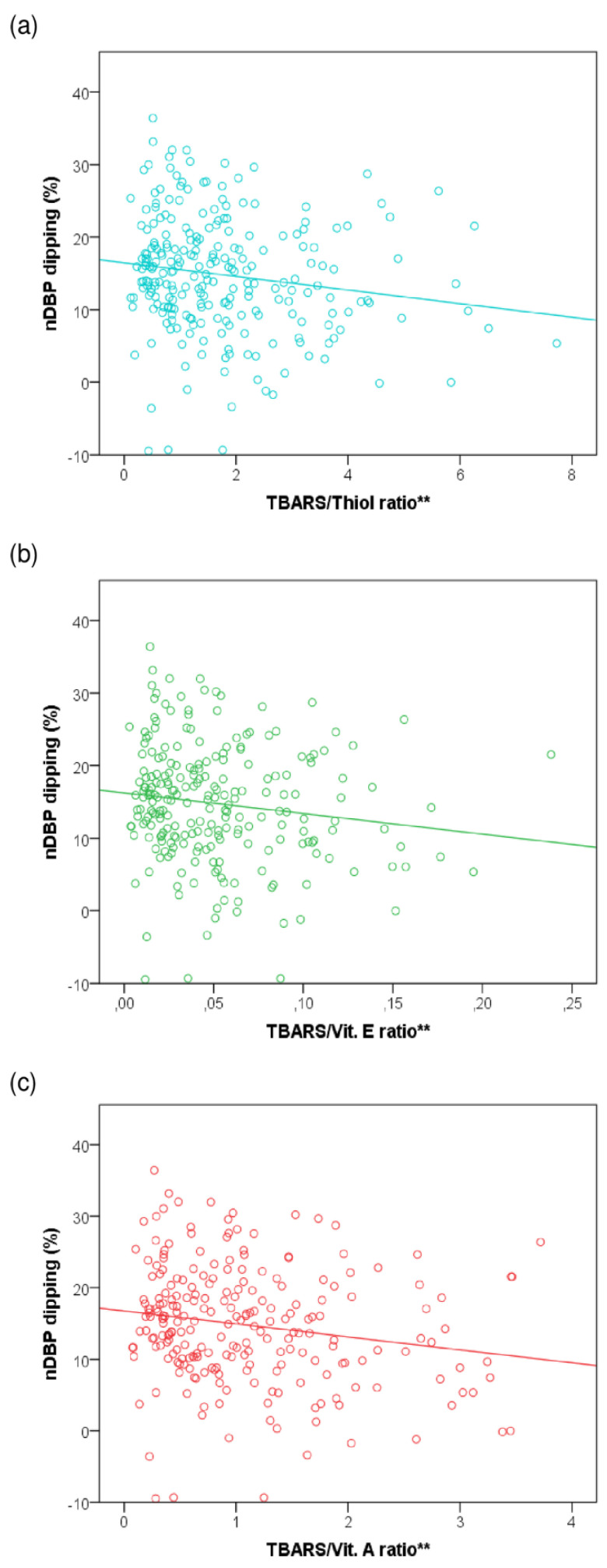
Correlations between nocturnal fall in diastolic blood pressure and oxidant/antioxidant ratios. The scatter plots represent the percentage of nDBP dipping according to the TBARS/Thiol, TBARS/Vitamin E, y TBARS/Vitamin A ratios. (**a**) TBARS/Thiol ratio (blue): 16.46 + (−0.94) × TBARS/Thiol. (**b**) TBARS/Vitamin E ratio (green): 16.24 + (−28.38) × TBARS/Vitamin E. (**c**) TBARS/Vitamin A ratio (red): 16.73 + (−1.80) × TBARS/Vitamin A. nDBP—nocturnal diastolic blood pressure. TBARS—thiobarbituric acid reactive substances. ** Refers to *p*-value lower than 0.05.

**Table 1 antioxidants-11-02430-t001:** Clinical and laboratory features: comparisons between patients with dipper and non-dipper diastolic blood pressure profiles.

Variables	Total Sample	DBP Profile		*p*-Value
	n = 248	Dipper n = 187	Non-Dipper n = 61	
Age (years) ^†^	56 (17)	55 (15)	58 (14)	0.045
Sex (women) ^‡^	138 (56)	108 (57)	30 (50)	0.300
Alcohol intake ^‡^	46 (18)	30 (16)	16 (26)	0.085
Former smokers ^‡^	92 (37)	74 (39)	18 (30)	0.220
BMI (Kg/m^2^) ^†^	28 (7)	28 (6)	29 (8)	0.066
WC (cm) ^†^	101 (18)	100 (19)	105 (21)	0.069
24-hSBP (mmHg) ^†^	125 (17)	123 (18)	128 (15)	0.035
24-hDBP (mmHg) ^†^	76 (12)	76 (12)	77 (14)	0.653
dSBP (mmHg) ^†^	129 (18)	128 (19)	130 (14)	0.690
nSBP (mmHg) ^†^	114 (18)	110 (16)	124 (15)	<0.001
dDBP (mmHg) ^†^	80 (14)	80 (13)	77 (14)	0.091
nDBP (mmHg) ^†^	68 (12)	66 (10)	75 (13)	<0.001
RAAS blockers ^‡^	140 (57)	101 (54)	39 (65)	0.178
Diuretics ^‡^	68 (27)	44 (24)	24 (40)	0.019
CCBs ^‡^	100 (42)	72 (40)	28 (47)	0.362
B-blockers ^‡^	40 (16)	28 (15)	12 (20)	0.420
Statins ^‡^	90 (36)	66 (35)	24 (40)	0.539
Compliant patients ^‡^	199 (81)	149 (80)	50 (83)	0.580
FPG (mg/dL) ^†^	99 (16)	99 (16)	102 (18)	0.312
Creatinine (mg/dL) ^†^	0.82 (0.3)	0.82 (0.3)	0.83 (0.3)	0.723
Uric acid (mg/dL) ^†^	5.0 (2.4)	4.9 (2.4)	5.3 (2.0)	0.110
Total proteins (g/dL) ^†^	7.2 (0.6)	7.2 (0.6)	7.3 (0.7)	0.178
TG (mg/dL) ^†^	92 (66)	89 (61)	110 (82)	0.004
TC (mg/dL) ^†^	188 (46)	187 (46)	190 (49)	0.732

BMI—body mass index. WC—waist circumference. SBP—systolic blood pressure. DBP—diastolic blood pressure. 24-hSBP—average 24 h SBP. 24-hDBP—average 24 h DBP. dSBP—average daytime SBP. nSBP—average nocturnal SBP. dDBP—average daytime DBP. nDBP—average nocturnal DBP. RAAS—renin–angiotensin–aldosterone system. CCBs—calcium channel blockers. FPG—fasting plasma glucose. TG—triglycerids. TC—total cholesterol. Cu—copper. Zn—zinc. Results expressed as ^†^ refer to median and interquartile range. Results expressed as ^‡^ refer to number and percentage.

**Table 2 antioxidants-11-02430-t002:** Binary logistic regression model based on the correlation between multiple oxidative stress markers and the presence of a non-dipper DBP profile.

Variables	B	*p*-Value	Exp(B)	CI95%	
				Inferior	Superior
Age (years)	0.042	0.035	1.043	1.003	1.085
nDBP (mmHg)	0.112	<0.001	1.119	1.070	1.169
Cu (µg/dL)	0.026	0.009	1.026	1.007	1.046
TBARS/Thiol ratio	0.538	<0.001	1.712	1.285	2.263

DBP—diastolic blood pressure. nDBP—nocturnal diastolic blood pressure. Cu—copper. TBARS—thiobarbituric acid reactive substances. Model summary: *p*-value (F-test) < 0.001, *p*-value (Hosmer and Lemeshow) = 0.269, −2LL = 176.2, R^2^ (Nagelkerke) = 0.373, Overall accuracy = 0.80.

## Data Availability

The data presented in this study are available on request from the corresponding author. In accordance with Article 18.4 of the Spanish Constitution and the Organic Law on Data Protection and Guarantee of Digital Rights (LOPDGDD) of 6 December 2018, the privacy and integrity of the individual will be protected at all times so anonymous data are available upon reasonable request.

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
