# Peer review of "Correlation between Blunted Nocturnal Decrease in Diastolic Blood Pressure and Oxidative Stress: An Observational Study"

_antioxidants, 2022, doi:10.3390/antiox11122430_

Round 1

Reviewer 1 Report

In the present cross sectional study, the Authors investigated the relationship between the nocturnal diastolic dipper status and several oxidative stress markers in 248 patients with hypertension.

They described that lower levels of reduced thiols, vitamin A, vitamin E with higher concentrations of TBARS in patients with a non dipper DBP pattern, while serum Cu levels were clinically higher in patients with a non dipper DBP profile. Moreover, they found a weak negative correlation between nDBP dipping and the TBARS/Thiol, TBARS/V itamin E and TBARS/Vitamin A ratios. At multivariate analysis, increased TBARS/Thiols ratio and serum Cu levels were the markers that best correlated with a non dipper DBP profile.

While the study is of some interest, there are some issues that raise concern over this manuscript

11)      Due to the description of findings in the Results section and data showed in Figure 2, the following sentences in the Abstract and in the Discussion section need to be corrected: “The nDBP decrease showed a weak positive correlation with reduced thiol, vitamin E and vitamin A levels and a weak negative correlation with Cu levels and the Cu/Zn ratio.” and “The results are summarized as follows: 1) nDBP decrease showed a positive correlation with reduced thiol, vitamin E and vitamin A levels and a negative correlation with Cu levels and the Cu/Zn ratio.”

22)     The following sentence in the Discussion section is not sustained by the findings and therefore should be erased.

Reviewer 2 Report

This is a thematically interesting study investigating the relationship between the decrease in night time DBP and selected antioxidant parameters in the plasma of patients with treated arterial hypertension. Although the results indicate a close relationship between insufficient DBP dipping at night and a lower levels of selected antioxidants, the presentation of the results is unclear in some cases.

Regarding the study, I have the following questions and comments related to the statistical analysis and presentation of the results:

·       In section 2.2. it is not stated by which methods the individual parameters were measured. These should be described or cited.

·       In the “Statistical analysis” section, it is stated that a “P-value of lower than 0.05 (P< 0.05) was considered for statistical significance.”  However, in the Results, differences with values of p < 0.1 are also specified. This is possible, but it is less commonly used and must be written and explain in the “Statistical analysis” section. However, I recommend to specify only the differences with p < 0.05.

In Table 1:

·       In the row “24-hDBP” is mean DPB equal to 76 in all groups. It is correct?

·       In the row “Compliant patients‡”  the numbers are 199 (247). The percentages are not given correctly. Please, correct.

·       Were creatinine levels really the same in all three groups?

Figure 1.

·       I don't understand Figure 1 exactly. Results are expressed as median and interquartile range (box?). But what are the whiskers in the figures? It is also not clear to me what the diamond symbols in the individual figures represent. Some diamonds are even covered with a graphic legend. These figures must be corrected and better explained in the legend.

·       In the legend to Fig. 1 is the abbreviation “DBP—Diastolic blood pressure”, but this variable is not shown in Fig. 1.

·       Also text " * Refers to P-value of lower than 0.1." is irrelevant for Figure 1.

Figure 3

·       Figure 3 shows the correlations of TBARS with 3 different oxidant/antioxidant ratios. Considering the scale of the x-axis, the TBARS/Vit E correlation is very unconvincing, rather it seems that there is no correlation. The authors should split Figure 3 into 3 separate figures so that an appropriate x-axis scale can be used for TBARS/Vit E.

 ·       Again the statement "* Refers to P-value of lower than 0.1. " is irrelevant for this Figure 3.

As written in manuscript,  binary logistic regression models for each redox marker was constructed. But this model is not specified in any closer way. Please, specify in more details how the multivariate analysis was performed or provide appropriate reference.

Line 330-  the term “arterial wall hyperpressure” is incorrect. Did you mean arterial hypertension? Please, explain and/or correct.

Round 2

Reviewer 2 Report

My comments were taken into consideration and the quality of the manuscript improved significantly. I have no other comments.